# Bioinformatic RNA-Seq Functional Profiling of the Tumor Suppressor Gene OPCML in Ovarian Cancers: The Multifunctional, Pleiotropic Impacts of Having Three Ig Domains

**DOI:** 10.3390/cimb47060405

**Published:** 2025-05-29

**Authors:** Adam G. Marsh, Franziska Görtler, Sassan Hafizi, Hani Gabra

**Affiliations:** 1Center for Computational Biology and Bioinformatics, University of Delaware, Newark, DE 19716, USA; 2Department of Oncology and Medical Physics, Haukeland University Hospital, 5009 Bergen, Norway; franziska.gortler@uib.no; 3School of Pharmacy & Biomedical Sciences, University of Portsmouth, Portsmouth PO1 2UP, UK; sassan.hafizi@port.ac.uk; 4Papyrus Therapeutics Inc., Philadelphia, PA 19104, USA

**Keywords:** OPCML, ovarian cancer, IgLON gene family, tumor suppressor genes, RTK suppression

## Abstract

The IgLON family of tumor suppressor genes (TSG) impact a variety of cellular processes involved in cancer and non-cancer biology. OPCML is a member of this family and its inactivation is an important control point in oncogenesis and tumor growth. Here, we analyze RNA-Seq expression ratios in ovarian cancers from The Cancer Genome Atlas (TCGA) (189 subjects at Stage III) to identify genes that exhibit a cooperative survival impact (via Kaplan–Meier survival curves) with OPCML expression. Using enrichment analyses, we reconstruct functional pathway impacts revealing interactions of OPCML, and then validate these in independent cohorts of ovarian cancer. These results emphasize the role of OPCML’s regulation of receptor tyrosine kinase (RTK) signaling pathways (PI3K/AKT and MEK/ERK) while identifying three new potential RTK transcriptomic linkages to KIT, TEK, and ROS1 in ovarian cancer. We show that other known extracellular signaling receptor ligands are also transcriptionally linked to OPCML. Several key genes were validated in GEO datasets, including KIT and TEK. Considering the range of OPCML impacts evident in our analyses on both external membrane interactions and cytosolic signal transduction, we expand the understanding of OPCML’s broad cellular influences, demonstrating a multi-functional, pleiotropic, tumor suppressor, in keeping with prior published studies of OPCML function.

## 1. Introduction

The IgLON family of tumor suppressor genes (TSG) were first described in expression studies of brain astrocyte cell cultures [1,2]. There are five protein family members with names reflecting their initial description in neuronal growth and differentiation: Opioid Binding and Cell Adhesion Molecule (OPCML), Neurotrimin (NTM), Neuronal Growth Regulator 1 (NEGR1), Limbic System Associated Membrane Protein (LSAMP), and an unnamed gene (IgLON5). However, early functional genomics screening of gene expression profiles in tumors revealed that IgLONs can function as TSGs; they demonstrated important roles in regulating cell growth, proliferation, and differentiation in a variety of tissues, including many types of tumors. In the early 2000s, OPCML was clearly shown to be silenced in the majority of ovarian cancer cases (via Knudson two-hit loss consisting of chromosomal loss (LOH) and somatic DNA methylation of two promoter CpG islands) [3,4,5,6].

Over the last two decades, much attention has been given to defining aspects of the mechanism through which OPCML controls receptor tyrosine kinase (RTK) signaling pathways through protein–protein interactions and plays an important natural physiological role in RTK regulation to prevent tumor growth [7,8,9]. OPCML has been shown to interact with at least six different RTKs; however, more still need to be tested [10]. In general, OPCML behaves as a chaperon to translocate RTKs into membrane “lipid raft microdomains”, where they become dephosphorylated and tagged for ubiquitin degradation. In healthy cells, constitutive expression of OPCML maintains an appropriate density of OPCML membrane protein dimers to allow physiological modulation of cell surface interactions and signal transduction pathways [9].

It is important to keep in mind that OPCML proteins in vivo are dimeric structures consisting of two monomer chains, each of which contains three immunoglobulin (Ig) domains anchored to the cell membrane via a glycosylphosphatidylinositol (GPI) moiety [11]. The dominant mechanism of inactivation is through LOH and somatic promoter island methylation; however, Birtley et al. identified 119 clinically reported OPCML mutations and investigated the impact of point mutations at 8 specific functionally relevant amino acid substitution sites (across all 3 Ig domains). This work demonstrated that the reported clinical mutations at these sites in different tumors inactivated OPCML’s tumor suppressor activity. Thus, each of the three Ig domains appear to be important to the overall functionality of the OPCML protein [9,12].

Ovarian cancer is a disease of tumor suppressor inactivation rather than a disease of oncogenic activating mutations. Tumor suppressors such as BRCA1, BRCA2, and p53 exemplify the nature of this disease. Ovarian cancer is also a disease of great unmet need, with poor outcomes for patients, and further insight into important tumor suppressors in ovarian cancer such as OPCML (inactivated in 82% of ovarian cancers) is an important justification for such research. The goal of this study is to look more deeply into identifying the multi-functional impacts of OPCML in ovarian cancer and the consequences of having three Ig domains in order to better understand the integrated mechanism of its pleiotropic functions.

## 2. Materials and Methods

Approach: We use RNA-Seq ratios (OPCML/gene_*i*_), where the resulting value essentially scales OPCML expression relative to the expression of each target gene_*i*_. Expression normalization to a reference gene (e.g., β-actin) is common in many differential gene expression studies. Our application of a ratio focuses the reference case on OPCML because we want to assess variations in expression of other RNA features relative to OPCML.

For each OPCML/gene_*i*_ ratio, the ovarian cancer cohort is divided into low and high ratio groups, using the median value as the threshold. Survival is then compared between the two groups of patients to identify genes that significantly impact patient health in relation to OPCML expression. Thus, we are using the phenotype “days_to_death” as the primary data to assess significance of the OPCML/gene_*i*_ ratios. To filter out false positives (in addition to conservative and empirical statistical thresholds employed), enriched functional analyses are used to identify pathways and processes that contain multiple gene_*i*_ members, not just one. From these functional gene sets, we then categorize the genes by cellular location to compile a multifunctional profile for OPCML impacts in ovarian cancer.

TCGA Study Cohort: Our subject cohorts utilize RNA-Seq data curated by the The Cancer Genome Atlas (TCGA) research network: https://www.cancer.gov/tcga (accessed on 15 January 2024). For high-grade serous ovarian cancer (TCGA-OV), inclusion criteria were stage IIIC tumors and integer “days_to_death” value (since diagnosis) greater than 0 and less than 3650 (10 years). These criteria resulted in an ovarian cancer cohort of 189 subjects after the removal of one individual with an extreme RNA-Seq value for OPCML transcript counts (16.2 standard deviation units above mean Z-score value).

A clear limitation of using TCGA cases is that these subjects include multiple sources of variance that are not uniformly controlled across all patients. This is because of the nature of aggregating subjects from diverse studies. The net effect is that this increased cohort variance just adds more background noise to any statistical analysis. The advantage of using TCGA cases is the shear number of subjects available in a cohort (here, 189 cases at stage IIIc). Thus, despite greater variances, the large number of patients provides a statistical power that enables researchers to uncover significant quantitative relationships, despite higher background noise.

Survival Analysis: Data download and initial handling used the TCGAbiolinks 2.31.2 R package for preprocessing and GC normalization [13,14]. These values were then transformed as $log_{2}\left(x\;+\;4\right)$ values to more uniformly scale the RNA-Seq ratios (OPCML/gene_*i*_). These ratios were calculated to scale OPCML expression to each gene feature in the RNA-Seq dataset (60,660 features). The linear scale-up by +4 above results in a minimum working value of 2.0 when normalized RNA-Seq values are near zero and ensures that the denominator in calculating the RNA-Seq ratios was never close to zero. This is a simple linear transformation commonly employed to reduce the impact of near-zero observations.

For each transcript feature, the ovarian cancer cohort was bifurcated into two groups by the median RNA-Seq ratio value (OPCML/gene_*i*_): (1) a “normal constitutive” ratio group (>median ratio; higher relative OPCML expression) and (2) a “suppressed or silenced” ratio group (≤median ratio; lower relative OPCML expression). Survival analysis of these cohort subgroups used the R packages survminer 0.5.0 and survival 3.8.3 to assess differences in univariate Kaplan–Meier survival curves (which utilize Log model *p*-values) and Cox hazard ratios (HR). All HR values are calculated as the relative survival difference for the “normal constitutive” OPCML expression subgroup. Our choice of univariate statistics is justified by our inclusion restriction of only patients with stage IIIC cancers and our objective of simply identifying the impacts of associative genes on survival. We are not developing further prognostic applications of these results in this paper.

A challenge to working with hazard ratios and applying probability distributions to assess significance is that HR distributions are neither symmetric nor balanced relative to the null hypothesis (Ho) assertion that the center of the HR distribution is 1.0 (i.e., no differential effect on survival). The lower tail of that distribution is limited to a range of 0 to 1. The upper tail can range from 1 to +*∞*. We applied a simple algebraic transform in R to establish a symmetrical distribution (Equation (Equation 1)):(1)$$HRsym=ifelse(coxHR<1,\left(-1\times \frac{1}{coxHR}+1\right),\left(coxHR-1\right))$$
Here the symmetrical HR value designated HRsym is derived from inverting the Cox HR when it is less than 1, and then adjusting the center of the null hypothesis distribution to assert HRsym = 0 when there is no difference in survival. We finally use a simple Z-score transform (HRz), yielding a distribution mean value of 0 (null Ho “no effect” center) with a standard deviation (sd) of 1. These processing steps are standard statistical approaches utilizing linear transformations to establish symmetric distributions to better define the significance threshold values in both the lower and upper tails of the distribution of the hazard ratio statistics.

HRz False Discovery Rates (FDR): The high number of survival curves being assessed for each gene ratio raises concerns of high false positive rates or FDR. In this study, Monte Carlo simulations were employed using randomized RNA-Seq data tables to essentially ascertain the range of likely result outcomes we would observe from our analysis pipeline if there were no functional linkages between genes and the only source of variance in the results was entirely due to simple random stochastic sampling events. Generating empirical null Ho distributions via Monte Carlo random sample simulations is a common practice in statistical modeling and has frequently been applied to analyses of high-dimensional genomics datasets [15,16].

The Monte Carlo models were executed with randomized RNA-Seq matrices. Survival “days_to_death” values among cohort subjects were also randomized. From the resulting distributions of HRz values, we established the lower 0.025 quantile limits (equivalent to α = 0.05 for a two-tailed distribution) for HRz values that would likely be observed (with 95% confidence) due to only random sampling variance across this high-dimensional space (number of subjects × number of features). These randomized HRz thresholds for statistical significance were then applied to the real-world OPCML *x *$gene_{i}$ data analyses to ensure that our acceptance of significant HRz values was not in excess of a controlled false positive rate of 0.025 on either tail of the distribution (net false discovery rate (FDR) < 0.05).

In addition, we applied a screening criterion for assessing significant HRz results requiring that the Kaplan–Meier Log model *p*-value of the survival curve for each OPCML/gene_*i*_ ratio stratification had to be at least 10% more significant than the Kaplan–Meier Log model *p*-value of a corresponding survival analysis with median stratification based only on the single *gene_i_* RNA-Seq values. In this way, we ensured that any final decision of a functional significant $Ratio_{i}$ relationship evidenced a higher significance (lower *p*-value) than the sole $gene_{i}$ expression by itself. As single gene RNA-Seq values, OPCML itself showed no significant survival differentiation in the ovarian cancer stage IIIc population under study.

Gene and Functional Enrichment Analyses: RNA-Seq features with significant HRz values were used to compile feature sets using custom R scripts. To better assess the functional significance of these gene sets, enrichment analyses used common curated databases: Gene Ontology (GO), Reactome (GSEA, Broad Inst., MIT), and Kyoto Encyclopedia of Genes and Genomes (KEGG). These different enrichment approaches yield different results and our approach here integrated these data streams into a functional snapshot of OPCML in ovarian cancer. The R packages clusterProfiler 4.11.0, enrichplot 1.22.0, and pathfindR 2.1.0 were used for executing analyses and visualizing results [17,18]. Most of the enriched results presented in this paper were generated by subnetwork association analysis of pathfindR 2.1.0 using a reference protein interaction network from the BioGRID database. For functional enrichment analyses, Benjamini–Hochberg FDR adjusted *p*-values were used at α = 0.050.

NCBI PubMed Central Citation Search: To guide identifying functions with high relevance to ovarian cancer, we utilized the PubMed database (US National Institute of Health, National Center for Biotechnology Information (NCBI)) to count the number of research publications over a 25-year period (1998 to 2023) relating each gene to ovarian cancer (accessed 5 September 2024). The PubMed database is open access and maintains a deep curation of citations and abstracts published in life science journals. A custom python script was used to iteratively query the database via a simple API interface for the association of all functionally significant ${gene}_{i}$ features in combination with the terms “AND+[ovarian cancer]+AND+1998:2023[PDAT]”. The return response was parsed for the citation counts, which were then saved for each ${gene}_{i}$ searched. An ovarian cancer relevance score (OVca) for each functional pathway was then calculated using these publication counts per gene. OVca scores represent the geometric mean of a function’s component genes that identified in the RNA-Seq survival analyses.

Validation: We utilized a de novo analysis of the RNA-Seq data for ovarian cancer cohorts from the Gene Expression Omnibus (GEO). The cohort GSE26712 was processed on our same code platform for the survival curve analyses as the TCGA-OV dataset. Other GSE datasets presented were evaluated using web-base tools KM Plotter (https://kmplot.com/analysis, accessed 12 February 2025) and ProgGenev2 (https://proggene.ccbb.indianapolis.iu.edu/, accessed 12 February 2025). Note that the GEO datasets utilized a different RNA-Seq platform than the TCGA-OV dataset used in this study.

## 3. Results

### 3.1. Novel Computational Approach

In this study, we assess the range of biological and cellular processes characterized and represented by transcripts that have synergistic associations impacting patient prognosis with RNA transcript expression levels of OPCML in ovarian cancer (stage IIIC, high grade serous carcinoma). Subject cohort and RNA-Seq data (189 subjects) were obtained from by The Cancer Genome Atlas (TCGA) Research Network. Our approach uses RNA-Seq ratios for OPCML normalized to each defined gene feature ((OPCML/${gene}_{i}$), where *i* ranges from feature number 1 to 60,660). Each ratio is used to bifurcate the subject cohort into a “normal constitutive” expression ratio group (>median ratio) and a “suppressed or silenced” expression ratio group (≤median ratio). We then test for survival (hazard ratios) comparing “normal constitutive” ratio cancer groups with “suppressed or silenced” expression ratio cancer groups to identify gene sets that exhibit a potential synergy on patient outcomes, relative to OPCML gene expression. Although this is an indirect association measurement, the phenotype being tested is patient overall survival, which is a robust and well defined outcome measured as “days_to_death” after primary diagnosis.

To filter out false positives, we employ conservative and empirically determined null Ho model thresholds. In addition, a filtering step via enriched functional analyses is used to identify pathways and processes that contain multiple gene_*i*_ members, not just a single significant gene. From these functional gene sets, we then categorize genes by cellular location to compile a functional snapshot of the scope of OPCML’s influence in ovarian cancer.

### 3.2. Survival Curve HR Analyses

The assessment of RNA-Seq data from an ovarian cancer cohort focuses on the potential survival impacts of each RNA feature (n = 60,660) in independent Kaplan–Meier (KM) survival curve analyses. For each RNA feature, the cohort was first stratified by median RNA-Seq expression to establish a “baseline” contribution to survival (positive or negative) for each gene or feature. In Figure 1, example KM curves are shown for OPCML and CKS1BP3 (CDC28 Protein Kinase Regulatory Subunit 1B Pseudogene 3), neither of which evidence any significant differential survival based on their solo expression levels. In subsequent rounds of RNA-Seq ratio testing (following), these baseline *p*-values and hazard ratios are used to gauge any statistically significant changes when assessing RNA-Seq ratios of OPCML/*gene_i_*.

The survival hazard ratio analyses are used to identify possible synergistic associations between OPCML and other genes that show differential survival when coupled with OPCML expression. In Figure 2a, an RNA-Seq ratio of OPCML and CKS1BP3 is shown as a distribution of values among cohort subjects. Median stratification divides the cohort into two subgroups with high vs low ratios. Although neither OPCML nor CKS1BP3 impact survival based on solo gene expression values (Figure 1), stratifying by the OPCML/CKS1BP3 ratio yields a significant difference in survival rate for the “High.Ratio” group (Cox HR = 0.657; model *p*-value = 0.0049; Figure 2b). Note that the hazard ratios are always calculated as the risk ratio of the High.Ratio group relative to the Low.Ratio group. Although no functional association is yet curated for this CKS1B pseudogene, the active expression of CKS1B has been recently linked to epithelial–mesenchymal transition (EMT) in gastric cancers [19], and the regulation of EMT by OPCML via direct interaction with AXL has been described in ovarian cancers [9].

The rationale for assessing RNA-Seq ratios with survival curves is that synergistic associations between genes can exist across large interaction networks and this approach can potentially reveal linkages that may be distantly connected, e.g., from the cell membrane surface to nuclear transcription factors. Overall, Figure 2 is presented as an example of the underlying survival curve analyses used to produce the HR values that underlie our functional assessment of OPCML. In the following analyses, our goal is to execute an integrative approach across all gene features to identify patterns that reveal potential functional shifts in cellular processes. When single gene impacts become reinforced by other genes in related pathways, then a possible linkage between OPCML and specific downstream functional shifts in cellular and molecular activities becomes more apparent.

To deal with skewed and biased raw Hazard Ratio distributions, HR values were scaled equivalently (Equation (Equation 1)) and then Z-transformed to ensure normal distributions around the null hypothesis assertion of no difference in survival (HRz = 0.0). Monte Carlo iterations of randomized RNA-Seq and survival data were used to empirically determine false discovery rate thresholds. The baseline false positive rate in these randomized RNA-Seq trials averaged 0.0522. In Figure 3, the distribution of transformed Z-scores (HRz) are shown, with the center of the distribution not statistically different from the null Ho assertion of no effect when HRz = 0 (*t*-test, mu = 0, *p* = n.s.). In addition, the tail quantile values at 0.025 and 0.975 are equivalent and the distribution is normally distributed (Shapiro–Wilk test, Ho: Wstat = 1.0, *p* = n.s.).

The distribution of HRz scores (Figure 3) has two prograding tails. The FDR thresholds are empirically determined and closely approximate the theoretical 95% core distribution boundaries for a standard normal Z-distribution. Functionally, these boundaries define two groups of genes: (a) OPCML^+^: HRz values < −2.0 identify OPCML/gene_*i*_ ratios with increased survival under constitutive OPCML expression relative to gene_*i*_ levels suggesting that lower levels of gene_*i*_ expression may be linked to increasing efficacy of OPCML pathways of tumor suppression (i.e., increased survival in the RNA-Seq High.Ratio group (see Figure 2b)); (b) ${\mathrm{GENE}}_{i}^{+}$: HRz values > +2.0 identify OPCML/gene_*i*_ ratios with increased when OPCML expression is lower and thus its pathways of tumor suppression are less active (i.e., increased survival in the RNA-Seq Low.Ratio group). Both of these groups (i.e., both tails of the HRz distribution) represent two distinct outcomes relative to OPCML expression levels.

### 3.3. Functional Enrichment Analyses

The collection of all significant gene features identified by survival RNA-Seq ratio HRz scores (Figure 3) were used for functional enrichment analyses. These were executed using the pathfindR 2.1.0 package to identify active sub-network structures based on a defined protein interaction network (PIN), the results of which were then used for enrichment analyses [17]. For this aspect of the work, four PIN database sources were used in serial combination with five pathway gene mapping databases (total independent enrichment analyses = 20). In Figure 4, the count of enriched functions identified in each combination of PIN x pathway are shown. The PIN analysis sets were fairly equivalent in count results. For subsequent analyses of functions, results from all five protein interaction networks were compiled together, filtering out multiple instances of the same enriched function by keeping the one with the highest reported *p*-value (more conservative value). This collection of unique entries resulted in a data table with a total of 242 significantly enriched identified functions, pathways, and processes.

Overall, hierarchical clustering of enriched functions by content of gene groups (OPCML^+^ vs ${\mathrm{GENE}}_{i}^{+}$; Figure 3) is an efficient way to organize and visualize functional groupings within a large set of results (242 enriched functions; see Section A.1, Figure A1)). However, hierarchical clustering of functions by their enriched gene composition does not reveal any large-scale distinct segregation of clusters based on the OPCML^+^ and ${\mathrm{GENE}}_{i}^{+}$ representation in function gene sets. There were 20 significant top-level clusters identified, with the presence of OPCML^+^ and ${\mathrm{GENE}}_{i}^{+}$ composition within those member functions not showing strong relationships in terms of gene composition. However, there are three exceptional clusters to this observation that are highlighted in Table 1.

In Table 1 (and Figure A1), cluster **a** is defined by 10 functions (6 OPCML^+^ and 4 ${\mathrm{GENE}}_{i}^{+}$ ) that have a high degree of gene overlap among them. This cluster involves Toll-like receptor (TLR) pathways. These entries are ranked by *p*-values and the top six have a >4× majority of OPCML^+^ genes. The four functions with more ${\mathrm{GENE}}_{i}^{+}$ composition form a discrete subgroup, but have sufficient overlap in enriched genes to be highly related in terms of pathway and process activities. In cluster **b**, there are 31 functions evenly split on the OPCML^+^ and ${\mathrm{GENE}}_{i}^{+}$ classifications. The four most significant functions in this cluster involve PI3K/AKT signaling in cancer (see cluster *b* asterisk Figure A1). We know that OPCML drives degradation of RTK signaling receptors [12] in cancers. This large cluster **b** aligns with how “constitutive normal” expression of OPCML contributes to increased survival in patients. It is interesting to note that when ranked by *p*-values in Table 1, the top ten functions are all in the >4× OPCML^+^ gene count class. Again, this is suggestive of a proximal cellular function by OPCML.

In cluster **c** (Table 1 and Figure A1), there are eight functions. In contrast to clusters **a** and **b**, seven of the eight are in the >4× OPCML^+^ gene count class, and the cluster break is at a high level in the tree (at branch P) and thus represents a distinctive functional grouping. In Table 1, this cluster focuses on external membrane cAMP signaling and connected pathways. Of note is the function “Calcitonin-like ligand receptors” (CLRs; Reactome 419812) with a ×Fold-enrichment of 16.2×, the highest in this study. CLRs have potent tumor micro-environment impacts in terms of stimulating both angiogenesis and cell proliferation [20]. The tumor suppressor activity of OPCML, particularly within the tumor micro-environment, is clearly antagonistic to the action of CLRs in tumor formation.

To further guide identification of functions with high relevance to ovarian cancer, we utilized the PubMed database (US National Institute of Health, National Center for Biotechnology Information (NCBI)) to count the number of research publications over a 25-year period (1998 to 2023), relating our gene sets to ovarian cancer. For every single ${gene}_{i}$ found in our enriched functions, a PubMed search was executed via API to count all articles where that ${gene}_{i}$ appeared in combination with the subject “ovarian cancer”. In Figure 5, the total ${gene}_{i}$ PubMed counts are shown for both OPCML^+^ and ${\mathrm{GENE}}_{i}^{+}$ groups. A primary assumption is that the genes with the highest number of publications in the field of ovarian cancer are likely to be those that are “known” to be relevant to the progression, severity, and treatment of these tumors. Firstly, although both gene groups are well represented in the ovarian cancer literature, the OPCML^+^ group has 2.4x more publication counts among the top 30 genes than the ${\mathrm{GENE}}_{i}^{+}$ group (2766 vs. 1164).

Secondly, the top six genes, each with more than 100 publications (KIT, EGF, STAR, GAL, FGF2, VIP), have known established roles in ovarian cancer. These genes have been studied and published because of their relevance to this cancer. Thirdly, these gene-level PubMed counts can be combined into an ovarian cancer relevancy score for each enriched function as a roll-up score (geometric average) from the constituent genes found in an enriched function. This scoring approach, calculated for all enriched functions identified, allows for a triage ranking of functions relative to their potential involvement with onset and progression of ovarian cancer.

Our “known” ovarian cancer relevancy score (OVca) was used to sort enriched functions and collapse the complexity and overlaps of the clustering relationships (Figure A1) into a ranked order for functions with >4× majority of OPCML^+^ genes and those with less than a 4× majority (i.e., more ${\mathrm{GENE}}_{i}^{+}$ component genes). Within any function cluster with more than four members, selection was limited to the top four functions by OVca score. Table 2 presents these top 30 ranked functions. Because of the volume of ovarian cancer research published involving constituent genes of these functions, the known relevance of these pathways to ovarian cancer is high. In Figure 5, KIT, EGF, and FGF2 are in the top five genes by publication count. In Table 2, functions with the highest OVca scores are those where KIT, EGF, and FGF2 were identified as constituent genes. For molecular oncologists, there are no surprises at the top of this table. However, the strength of this table is that all results are based upon a statistical linkage between OPCML/${gene}_{i}$ RNA-Seq ratios and an increase in survival observed in patients with “normal constitutive” levels of OPCML expression with conversely statistically significant adverse survival in those with low OPCML/${gene}_{i}$ ratios. OPCML is linked to all the member genes of these functions in some fashion. Thus, the cellular-level impacts of OPCML are far reaching through many mechanisms of tumor progression and disease severity.

### 3.4. Validation

We subjected the key OPCML RNA-Seq ratios identified as significant for survival difference from this analysis (TCGA-OV cohort) to validation from independent cohorts of ovarian cancer (GEO/GSE) (see Table A1). No two patient cohorts are identical. Usually, they are not very close in terms of the sources of variance within each. The inclusion/exclusion criteria in a study are tuned to the goals of the research team(s) and the population of patients available near enrollment centers. We selected the TCGA OV cohort because we could stratify the almost 600 subjects down to 189 subjects with common diagnosis: stage III ovarian cancer. The unifying phenotype of our study is that all subjects have the same ovarian cancer development state.

To assess the extensibility to other ovarian cancer cases (without controlling for stage, age, race, method, prior treatments, etc.) we selected key features identified from Table 2 and Figure 6. The goal was not to repeat the study in another cohort set, but to assess whether the features we identified specifically in TCGA Stage III cases are likely to appear in other studies of different ovarian cancer cases. This supervised analysis confirmed that many of these transcripts were significant in the validation analysis cohorts. Positive hits are listed in Table A1 and represent about half of the total gene_*i*_ targets that we pursued for validation. These results underscore the validity of our approach for uncovering molecules potentially involved in the clinical biology of OPCML. The subject heterogeneity between different studies (TCGA vs GSE) is high given that the former uses Illumina TruSeek method (∼60 k features), while the latter uses Affymetrix methods (∼22k features) for RNA-Seq measurements. Despite such quantitative differences, we were able to verify common features between both, including the associations with KIT and TEK. The orthogonal nature of validation is a strong indicator of a functional impact of key associations found. We are confident that our results are directly applicable to ovarian cancer stage III cases.

## 4. Discussion

### 4.1. RTKs, Ligands, and Signaling

One of the principal functions of OPCML as a tumor suppressor protein is to inactivate receptor tyrosine kinases (RTKs) by binding and chaperoning RTKs to the lipid raft followed by dephosphorylation of the activated RTK and subsequently driving endocytosis, leading to degradation of that RTK. In ovarian cancers, OPCML drives inactivation and degradation of the RTKs HER2, HER4, FGFR1, FGFR3, EPHA2, and AXL [10], and the specificity of this network is underscored by identification of a group of RTKs that are unaffected by OPCML. There may, however, be more RTKs impacted by OPCML to be described with further research. AXL is a good example of this inactivation/degradation pathway. When AXL is complexed with its ligand GAS6 in ovarian cancer cells, OPCML chaperones the AXL-GAS6 complex into external membrane lipid-raft domains where the complex becomes dephosphorylated through apposition to a raft restricted phosphatase and then targeted for ubiquitin-dependent degradation [9,12]. In the last decade, there has been a growing recognition of the important role that AXL and other RTKs play in ovarian cancers and their role in signaling through PI3K/AKT and MEK/ERK signaling pathways. By driving the dephosphorylation of RTKs within lipid raft domains, OPCML may prevent prevent low avidity aggregation of RTK species that leads to high probability initiation and maintenance of downstream signaling, which is crucial for the spectrum of pleiotropic functions exhibited by RTK-driven cancers. Thus, we know that OPCML plays a critical role in negatively regulating cancer initiation and progression through its interaction with RTKs at their site of action, the external leaflet of the cell membrane of ovarian cells.

In this work, we demonstrate that in addition to this primary effect on RTKs, OPCML can influence a broader range of cellular activities that reinforce the potency of its tumor suppression effects. Beyond OPCML’s chaperoning effects on RTKs into cell-membrane lipid rafts, the analyses reported here regarding functional pathway enrichment (Table 2) emphasize deeper secondary association impacts of OPCML with other cellular processes that also impact tumor growth.

In Figure 6, a summary set of gene relationships and functional associations are compiled to illustrate the transcriptomic linkages between OPCML and genes that control broad functional aspects of tumor growth. These analyses are all based on RNA-Seq transcriptome data relative to overall survival in ovarian cancer patients. Here we see a downstream tumor suppressor role for OPCML beyond the direct interaction with membrane RTKs previously reported, and validated orthogonally by previously reported pathways in independent ovarian cancer cohorts. The RNA-Seq ratio analyses reveal significant linkages with genes that span a range of tumor growth activities. Overall, the proteins shown in Figure 6 are organized by cellular location and general function groupings: (1) known RTK signaling ligands, (2) extracellular matrix proteins, (3) other extracellular ligands, (4) other membrane receptors and proteins, (5) a few intracellular proteins involved with signaling, and (6) two transcription factors and one nuclear protein. Note that the circle symbols in Figure 6 with an asterisk indicate OPCML/${\mathrm{GENE}}_{i}^{+}$ ratios that have a direct and significant survival hazard ratio (HRz) for patients.

Highlighted in Figure 6 are the RTKs KIT, ROS1, and TEK that are significantly associated with OPCML RNA-Seq ratios in this study. These associations have not as yet been reported in any direct experimental studies. However, patient survival HRz scores reveal a new and significant RNA-Seq level linkage between OPCML and KIT, while TEK and ROS1 are prominent in several of the enriched function pathway results. Note that in Figure 5, there are just over 1000 literature citations of KIT in ovarian cancer in the last 25 years.

A key observation in Figure 6 is that there is a diverse range of transcriptomic links between OPCML and other proteins, mainly across the extracellular matrix and cell membrane compartments. In this RNA-Seq analysis, we see far broader and more downstream associations at a transcriptional level (*c.f.* direct protein–protein interaction studies), indicative of the multifunctional “downstream-ripple” effect of OPCML expression. The enumeration of functionally enriched pathways (Table 2) demonstrates a wide diversity of functions at granular resolution. In contrast, the level of proteins that impact tumor growth processes in Figure 6 reveal how many of these granular pathways align with OPCML’s general cellular activities as a tumor suppressor gene.

### 4.2. OPCML: Multifunctional Tumor Suppressor

Given that 49 out of 58 proteins (84%) in Figure 6 are operative externally to the cell or within the cell membrane, the site of action of OPCML, we conclude that OPCML has multiple functional interactions with transmembrane proteins that extend from the extracellular side of the membrane to the cytosolic side of the membrane proximal to PI3K/AKT and MEK/ERK signal transduction pathways. In this study, there are 29 extracellular/membrane functional proteins with linkage to OPCML in contrast to only six intracellular proteins. The impact of OPCML expression in ovarian cancer cells supports strong relationships to external, microenvironment-facing signal networks.

Alongside the regulation of classic outside-in signaling via cell surface receptors, extracellular proteins can impact three important processes in tumor growth: interactions between tumor cells, interactions between tumor and immune system cells, and modulation of tumor microenvironments. In Figure 6, we have identified 15 ligands and 8 proteins in the extracellular space that are important players in these processes. The ligands are mostly chemokines and cytokines (POMC, SHH, IL12A, IL4, CCL11, CXCL6, IL5, CXCL1, LEP, and TNFSF11) that influence immune system activity and regulate tissue inflammatory responses. OPCML’s spatial relationship with the extracellular compartment with its three Ig domains allows for the opportunity to interact with other proteins and small molecules in this space. As a result, we see transcriptional linkages between some of these ligands and OPCML. Other proteins present here are involved in matrix remodeling, which can alter tumor invasiveness (MMP13, S100A8, S100A9, and ELN).

In the membrane compartment there are also associations with G protein-coupled receptors (GPCRs: PTGIR, OPRM1, ADRA1A, HTR2A, HTR2B, FPR2, and NMUR2). We interpret these linkages as potential feedback loops between GPCRs and OPCML to influence OPCML’s cell-surface adhesion molecule role. In addition to these GPCRs, other significant signaling membrane proteins that are impacted by OPCML are FGR, CD36, and NGFR.

This RNA-level association network is obviously different from previously demonstrated direct protein–protein interaction networks for OPCML, although this study does present an orthogonal context for validation of prior published observations. We observe a reduced mortality risk when OPCML expression levels are in a normal, constitutive expression range with associated expression of other signaling pathway genes that are below the cohort median. This suggests that the seesaw balance of down- vs. up-regulation of cell signaling in cancers extends across many parallel functional pathways.

### 4.3. Translational Application of Results

Therapeutics: There have been a range of studies looking at OPCML restoration as a therapeutic agent to restore its natural tumor suppressor functions when those activities are lost due to OPCML gene silencing. In designing a therapeutic strategy to enhance an IgLON family TSG activity in tumor cells, two options have been considered: (a) whole-protein replacement therapy for OPCML [21,22,23] or (b) a peptide fragment derived from the first Ig domain for NEGR1 [24,25].

OPCML is unique in exerting its tumor suppressive function on the external side of the cell membrane through RTK modulation in lipid raft microdomains. Thus, OPCML operates extracellularly as a GPI-anchored protein and its post-translational mechanism is clearly distinct from that of other defined tumor suppressors, including genes that function through transcriptional repression (e.g., TP53), chromatin remodelling (e.g., ARID1A), or the cytoplasmic protein PTEN which regulates AKT signaling.

This membrane-based mechanism has critical implications for targeted therapy development. Firstly, unlike intracellular tumor suppressors that are often difficult to replace pharmacologically, OPCML’s extracellular localization offers a more accessible therapeutic targeting approach for drug delivery. Secondly, it highlights the therapeutic potential of restoring or mimicking OPCML function, such as through recombinant OPCML protein delivery. Finally, because OPCML can disrupt RTK dimerization and sensitize tumors to RTK inhibitors, its presence or absence could serve as a predictive biomarker for treatment response.

Experimental studies of OPCML interactions with AXL and HER2 indicate that both monotherapy with OPCML protein therapeutic and also combination therapies with OPCML and RTK inhibitors could be effective treatments. In HER2-expressing ovarian and breast cancer cell lines, responses to both lapatinib (anti-HER2 small molecule therapeutic) and erlotinib (anti-EGFR) were enhanced with exogenous OPCML [8]. In ovarian cancer cell lines, OPCML enhanced the effect of the the AXL inhibitor bemcentinib both in vitro and in vivo [9]. Also, in a type of liver cancer (cholangiocarcinomas), OPCML enhanced the effect of bemcentinib in AXL-expressing cell lines [26]. In looking over the landscape of OPCML linkages in Figure 6, there is a broad range of OPCML “touch-points” that could be useful to exploit for therapeutic efficacy. As in the above examples, mapping current drugs and their targets to this landscape could identify potential synergies via OPCML therapies that could synergise with these agents.

As a master regulator of RTK activity through non-canonical, extracellular mechanisms, OPCML offers several promising therapeutic avenues to combat drug resistance in cancer, particularly redundant by-pass overlaps between RTKs. These include its promotion of RTK degradation (e.g., HER2, FGFR1) or facilitating phosphatase-mediated deactivation (e.g., AXL). In these ways, OPCML restores balance to hyperactive signaling pathways commonly implicated in primary oncogenic signaling and also signaling associated with targeted therapy resistance. One strategy involves using recombinant OPCML protein therapeutically to reintroduce its tumor-suppressive effects through binding the ECD of RTKs at the external leaflet of the cell membrane within the lipid raft. Preclinical studies in ovarian and breast cancer models have demonstrated that treatment with recombinant OPCML reduces tumor growth and enhances sensitivity to small molecule RTK inhibition. Similarly, OPCML’s repression of AXL, also in synergy with selective inhibitors, can more effectively target EMT in cancer cells, especially as an approach to overcome resistance to prior chemotherapies. This includes targeting those cells’ enhanced invasiveness and metastasis, two hallmarks of aggressive, drug-resistant tumors.

Biomarkers: Another view of the OPCML landscape in Figure 6 would be to assess these gene/protein targets related to OPCML for functional biomarker utilization. Knowing which pathways through this network are active in a tumor could inform therapeutic efficacy of combination therapies given a patient’s specific tumor molecular phenotype. A recent review of the importance of genomic-level biomarkers for characterizing ovarian cancers presents a strong argument for the essential need to better stratify for this cancer among patients [27]. The landscape in Figure 6 illustrates a network of associations, and understanding gene-level sequence variants within that network would be highly relevant to classifying ovarian cancers and initiating the development of additional biomarkers. A further extension of such biomarker development would be an application toward earlier cancer diagnoses, and selection of particular therapeutic approaches that are extremely important in gynecological cancers where most diagnoses are made only after symptoms are evident [28].

Limitations of Approach: The dependence on large cohorts for statistical resolution (>100 subjects), consistent survival data for each subject and matched RNA-Seq chip platform data narrow the scope of application primarily to large curated databases. Although this analysis is excellent at picking up the signal of genes related to OPCML pathway-dependent tumor suppression (see Figure 3, OPCML^+^ genes), it leaves a gap in understanding of the functional relationships of the ${\mathrm{GENE}}_{i}^{+}$ members where a survival increase (albeit just a greater number of “days_to_death”) is apparent when OPCML expression is low.

Epigenetic silencing of OPCML is prevalent in ovarian cancers, with 82% of cases evidencing promoter methylation at two separate CpG dinucleotides upstream of the OPCML transcription start site [3]. In the other fraction of ovarian cancer cases where OPCML silencing is not involved, there are other drivers and controls of tumor formation that are independent of the status of OPCML expression. A better understanding of the ${\mathrm{GENE}}_{i}^{+}$ group members could provide information about these tumors, but this study is focused on the majority of tumors where OPCML driven turnover of RTKs is absent.

### 4.4. Summary

Overall, the results highlight, confirm, and underscore the role of OPCML’s regulation of RTK signaling and the downstream signaling pathways of PI3K/AKT and MEK/ERK, and also identify three new potential RTK linkages to KIT, TEK, and potentially ROS1 in ovarian cancers in the context of OPCML regulation. Functional pathway enrichment analyses indicate a broad array of secondary associations of OPCML that translate to survival benefits for patients when OPCML gene expression is near normal constitutive levels. The high degree of independent validation of our observation underscores the methodological power of our in silico approach and encourages our work to extend into other IgLONs involved in cancer. OPCML is a multi-functional, pleiotropic, tumor suppressor protein that can inhibit tumor formation via multiple, distinct primary and secondary pathways that significantly impact patient survival outcomes, consistent with its known functional biology.

## Figures and Tables

**Figure 1 cimb-47-00405-f001:**
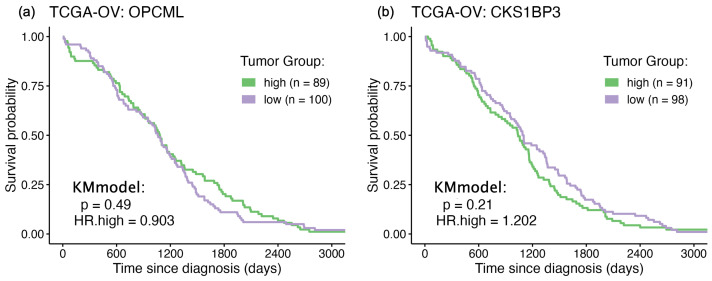
Kaplan–Meier survival curves: solo genes. The ovarian cancer cohort is median stratified into two subgroups by RNA-Seq expression (gene counts) and differential survival rates are assessed via Kaplan–Meier curves (KMmodel) and Cox hazard ratios (HR). Some examples are as follows: (**a**) OPCML median ratio stratification, *p*-value = n.s.; (**b**) CKS1BP3 median ratio stratification, *p*-value = n.s. (CDC28 Protein Kinase Regulatory Subunit 1B Pseudogene 3).

**Figure 2 cimb-47-00405-f002:**
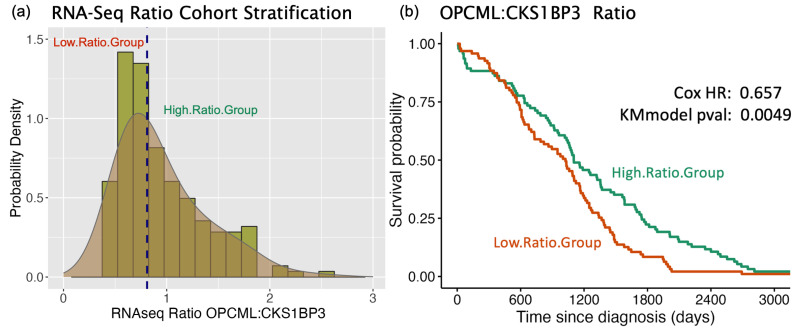
Kaplan–Meier survival curves: RNA-Seq ratios. As an example, ovarian cancer subjects are bifurcated by the ratio of OPCML expression normalized to CKS1BP3. (**a**) Probability density for OPCML/CSK1BP3 RNA-Seq ratios. Dashed blue line represents median value separating High.Ratio from Low.Ratio subgroups. (**b**) Survival curve comparison between subgroups (Kaplan–Meier model *p* = 0.0049).

**Figure 3 cimb-47-00405-f003:**
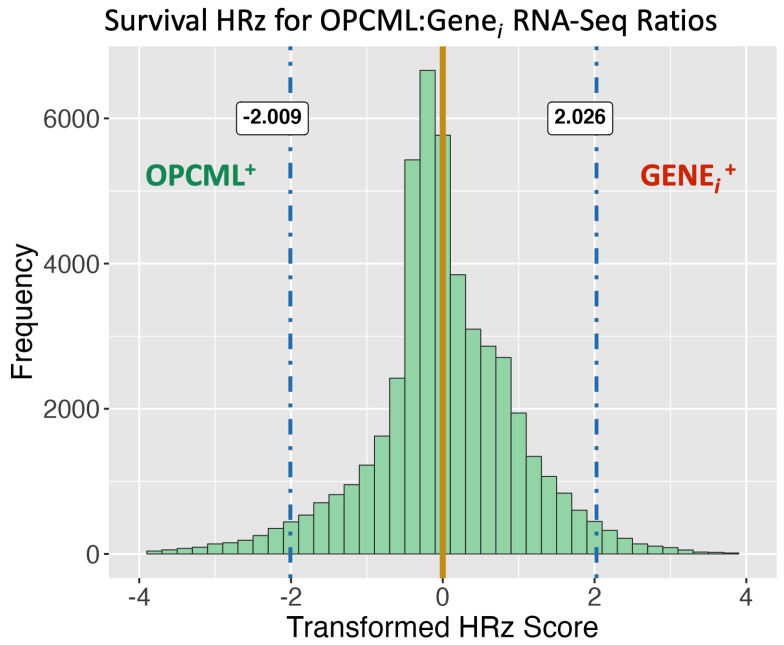
Transformed Cox hazard ratio distributions: HRz. Z−score distribution of survival Cox HR values (HRz) in ovarian cancer subjects. The null hypothesis value of no effect (Ho: HRz = 0.0) is shown along with the Monte Carlo False Discovery Rate quantile values for 0.025 and 0.975 tails (blue dashed lines). These FDR thresholds define two groups of genes: OPCML^+^ and ${\mathrm{GENE}}_{i}^{+}$.

**Figure 4 cimb-47-00405-f004:**
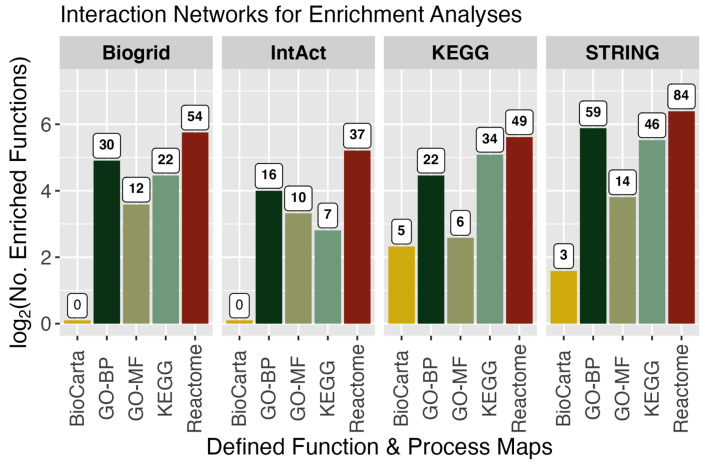
Functional network enrichment analyses. Subnetwork association analyses for functional enrichment were executed using different protein interaction network databases and pathway gene map databases. Four protein association networks were applied to the analysis of pathways defined in five database sources (total independent analyses executed = 20). Enriched function counts are plotted as $log_{2}\left(n\right)$, with the integer count included as a label.

**Figure 5 cimb-47-00405-f005:**
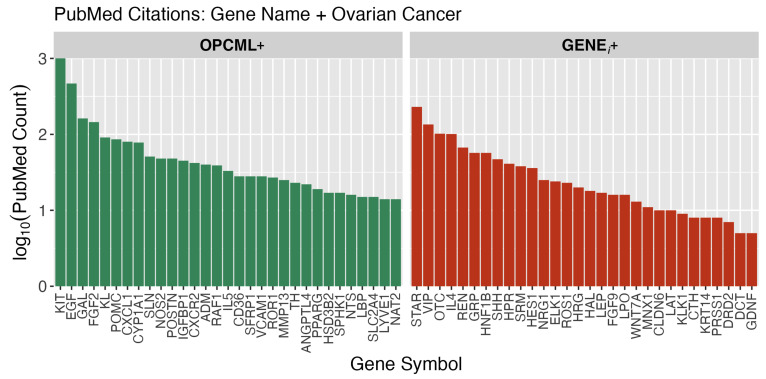
PubMed counts for genes associated with ovarian cancer. Each gene with a significant HRz value that was also identified in at least one enriched function was used in a literature search of NCBI’s PubMed database with the inclusion term “AND+[ovarian cancer]”. The genes are divided into OPCML^+^ and ${\mathrm{GENE}}_{i}^{+}$ groups based on their HRz values. The top 30 genes in each group ranked by PubMed citation counts are shown.

**Figure 6 cimb-47-00405-f006:**
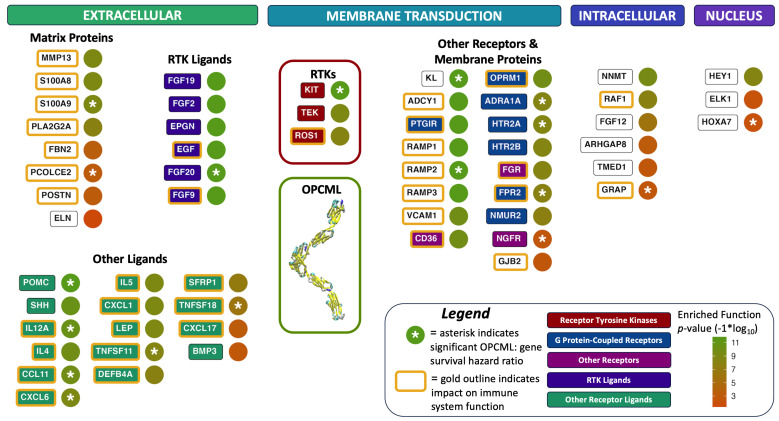
OPCML is transcriptionally linked to many proteins that impact tumor growth. Protein Genes with RNA-Seq values that identified them as contributing to enriched functional pathways (Table 2) are highlighted here, relative to their cellular location. The interaction of OPCML with RTKs is one of the primary effects OPCML exerts as a tumor suppressor gene. However, there are other secondary linkages via transcription RNA-Seq data that reveal a wide range of coordinated effects OPCML has on other processes related to tumor growth. The *p*-value color scale indicates the significance of enriched functions in which genes were found to be an expressed member.

**Table 1 cimb-47-00405-t001:** Example cluster groups of related functions: Three example branches from the dendrogram in Figure A1. The cluster labels “a”, “b”, and “c” refer to the groups highlighted in Figure A1 by the light-blue boxes.

ID	*p*-Value	×Fold	Survival Group	Description
a: cluster 16 (*n* = 10)				
R5602498	1.51 $\times \;10^{-9}$	6.48	OPCML+	MyD88 deficiency (TLR2/4)
R5603041	1.51 $\times \;10^{-9}$	6.07	OPCML+	IRAK4 deficiency (TLR2/4)
R5686938	3.58 $\times \;10^{-9}$	4.86	OPCML+	Regulation of TLR by endogenous ligand
R166016	1.18 $\times \;10^{-8}$	1.71	OPCML+	Toll-like receptor 4 (TLR4) Cascade
R5602358	1.64 $\times \;10^{-8}$	3.24	OPCML+	Diseases associated with the TLR signaling cascade
R5260271	1.64 $\times \;10^{-8}$	3.24	OPCML+	Diseases of immune system
R181438	9.19 $\times \;10^{-8}$	1.74	GENE+	Toll-like receptor 2 (TLR2) Cascade
R168179	9.19 $\times \;10^{-8}$	1.74	GENE+	Toll-like receptor TLR1:TLR2 Cascade
R166058	9.19 $\times \;10^{-8}$	1.78	GENE+	MyD88:MAL(TIRAP) cascade initiated on plasma membrane
R168188	9.19 $\times \;10^{-8}$	1.78	GENE+	Toll-like receptor TLR6:TLR2 cascade
b: cluster 1 (*n* = 31; top 10 shown)				
R2219530	1.41 $\times \;10^{-12}$	2.52	OPCML+	Constitutive signaling by aberrant PI3K in cancer
R2219528	4.14 $\times \;10^{-12}$	1.87	OPCML+	PI3K/AKT signaling in cancer
R6811558	4.14 $\times \;10^{-12}$	1.87	OPCML+	PI5P, PP2A, and IER3 regulate PI3K/AKT signaling
R199418	4.96 $\times \;10^{-12}$	1.75	OPCML+	Negative regulation of the PI3K/AKT network
R109704	5.83 $\times \;10^{-12}$	2.76	OPCML+	PI3K cascade
R112399	8.44 $\times \;10^{-12}$	2.53	OPCML+	IRS-mediated signalling
R5654219	9.69 $\times \;10^{-12}$	6.07	GENE+	Phospholipase C-mediated cascade: FGFR1
R2428928	9.69 $\times \;10^{-12}$	2.34	OPCML+	IRS-related events triggered by IGF1R
R2428924	9.69 $\times \;10^{-12}$	2.29	OPCML+	IGF1R signaling cascade
R74751	9.69 $\times \;10^{-12}$	2.25	OPCML+	Insulin receptor signalling cascade
c: cluster 6 (*n* = 8)				
R418555	1.69 $\times \;10^{-12}$	2.35	OPCML+	G alpha (s) signalling events
R419812	1.84 $\times \;10^{-12}$	16.2	OPCML+	Calcitonin-like ligand receptors
R373080	2.96 $\times \;10^{-11}$	4.0	OPCML+	Class B/2 (Secretin family receptors)
hsa04270	8.68 $\times \;10^{-9}$	3.04	OPCML+	Vascular smooth muscle contraction
G0007189	1.49 $\times \;10^{-8}$	4.77	OPCML+	adenylate cyclase-activating G protein-coupled receptor
G0031623	8.13 $\times \;10^{-6}$	3.74	OPCML+	receptor internalization
G0015031	0.0007	5.28	GENE+	external membrane protein transport
G0006816	0.0106	2.31	OPCML+	calcium ion transport

**Table 2 cimb-47-00405-t002:** OPCML+ majority enriched functions with ovarian cancer relevance score. The top 30 enriched functions ranked by known relevance to ovarian cancer via PubMed citation counts. The top three (max) constituent genes in both OPCML^+^ and ${\mathrm{GENE}}_{i}^{+}$ groups are listed. These gene groups are defined by the HRz ratios and are noted here as “OPCML Pathway” dependent vs. independent in terms of tumor suppressor activity.

				OPCML Pathway		
				**Dependent**	**Independent**	
**ID**	* **p** * **-Value**	**×Fold**	**OVca**	**Top 3 OPCML^+^ **	**Top 3 ${\mathrm{GENE}}_{i}^{+}$ **	**Description**
R2219530	1.41 $\times \;10^{-12}$	2.52	323.3	KIT, EGF, FGF2	FGF9	Constitutive signaling by aberrant PI3K in cancer
R2219528	4.14 $\times \;10^{-12}$	1.87	323.3	KIT, EGF, FGF2	FGF9	PI3K/AKT signaling in Cancer
R6811558	4.14 $\times \;10^{-12}$	1.87	323.3	KIT, EGF, FGF2	FGF9	PI5P, PP2A, and IER3 regulate PI3K/AKT signaling
R199418	4.96 $\times \;10^{-12}$	1.75	323.3	KIT, EGF, FGF2	FGF9	Negative regulation of the PI3K/AKT network
hsa04640	1.69 $\times \;10^{-5}$	1.91	271.0	KIT, IL5, CD36	IL4	Hematopoietic cell lineage
R6785807	1.42 $\times \;10^{-10}$	3.01	257.5	FGF2, POMC, NOS2	IL4	Interleukin-4 and Interleukin-13 signaling
hsa04072	0.0029	1.76	255.9	KIT, EGF, CXCR2	DGKK, GRM6	Phospholipase D signaling pathway
G0051897	1.74 $\times \;10^{-5}$	2.31	221.8	KIT, EGF, FGF2	LEP, ERFE	Positive regulation of phosphatidylinositol 3-kinase/protein kinase B signal transduction
hsa04913	0.0001	1.98	201.9	STAR, CYP1A1, HSD3B2		Ovarian steroidogenesis
G0006954	1.49 $\times \;10^{-5}$	1.87	197.3	KIT, CXCR2, ADM		Inflammatory response
R375276	1.86 $\times \;10^{-8}$	2.69	173.1	GAL, POMC, CXCL1	GRP, GPR37L1, PENK	Peptide ligand-binding receptors
hsa04927	1.10 $\times \;10^{-6}$	2.25	150.8	STAR, POMC, HSD3B2		Cortisol synthesis and secretion
hsa04657	1.07 $\times \;10^{-9}$	1.99	150.7	CXCL1, IL5, MMP13	IL4	IL-17 signaling pathway
G0001938	1.74 $\times \;10^{-5}$	2.64	127.9	EGF, FGF2, CCL11		Positive regulation of endothelial cell proliferation
hsa04020	2.58 $\times \;10^{-8}$	2.44	122.9	EGF, FGF2, SLN	FGF9, GDNF, SMIM6	Calcium signaling pathway
G0042531	0.0002	3.28	122.6	KIT, HES1, IL12A	IL4	Positive regulation of tyrosine phosphorylation of STAT protein
hsa04060	0.0016	1.77	109.1	CXCL1, CXCR2, IL5	IL4, LEP	Cytokine-cytokine receptor interaction
G0043406	0.0017	2.92	96.7	EGF, FGF2, TNFSF11		Positive regulation of MAP kinase activity
hsa04979	0.0002	1.98	95.6	STAR, CD36, ANGPTL4		Cholesterol metabolism
R114608	0.0077	1.74	93.8	EGF, CD36, SELP	HRG	Platelet degranulation
hsa04064	0.0001	2.0	91.9	CXCL1, VCAM1, LBP	LAT	NF-kappa B signaling pathway
G0048018	0.0133	2.92	88.6	EGF, NTS, CCL11	WNT7A	Receptor ligand activity
G0006935	1.01 $\times \;10^{-8}$	2.74	85.2	FGF2, CXCL1, CXCR2		Chemotaxis
R381340	1.24 $\times \;10^{-9}$	2.13	84.0	CD36, ANGPTL4, PPARG	LEP	Transcriptional regulation of white adipocyte differentiation
R9843745	3.30 $\times \;10^{-9}$	1.85	84.0	CD36, ANGPTL4, PPARG	LEP	Adipogenesis
G0051092	6.92 $\times \;10^{-7}$	2.28	83.6	ROR1, SPHK1, NTS	CRNN	Positive regulation of NF-kappaB transcription factor activity
G0001525	0.0058	2.43	77.5	EGF, HEY1, HOXA7	LEP	Angiogenesis
G0008201	0.0009	3.99	76.2	POSTN, SFRP1, LTF	HRG, AOC1	Heparin binding
G0004713	5.10 $\times \;10^{-9}$	2.81	75.3	KIT, FGR, BTK	ROS1	Protein tyrosine kinase activity
G0070374	0.0008	2.54	59.0	FGF2, CD36, TNFSF11	ARHGAP8	Positive regulation of ERK1 and ERK2 cascade

## Data Availability

All raw data used in this study are publicly available from The Cancer Genome Atlas (TCGA) Research Network: https://www.cancer.gov/tcga (accessed on 15 January 2024). Results from this project for RNA-Seq ratio survival curves and enriched function analyses are compiled into separate data tables available on FigShare under an MIT User License (DOI: 10.6084/m9.figshare.c.7588115).

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
