# Peer review of "Bioinformatic RNA-Seq Functional Profiling of the Tumor Suppressor Gene OPCML in Ovarian Cancers: The Multifunctional, Pleiotropic Impacts of Having Three Ig Domains"

_cimb, 2025, doi:10.3390/cimb47060405_

Round 1

Reviewer 1 Report

Comments and Suggestions for Authors

This manuscript presented a new way to use the RNASeq data for cancer prognosis analysis.  Based on the TCGA ovarian  Stage IIIC tumor samples, for each gene GENEi , the ratio OPCML/GENEi  was computed for each sample.  The Cox model was used to estimate the HR of  OPCML/GENEi .  The HRs were transformed to HRz values. The Monte Carlo randomized models were used to obtain the empirical null H0 distribution of HRz to examine the significance of observed HRz values.  The quantile values were used as thresholds to find two significant gene sets: OPCML+ and GENEi+. The independent datasets were used to validate some genes in the two gene sets. The PubMed Counts for genes associated with ovarian cancer were generated to show whether the genes in the two sets were relevant to ovarian cancer. The results are interesting.

Some major comments:

  1. The two gene sets were defined by OPCML+ = { Genei with HRz < -2.009} and

GENEi+ = { Genei with HRz  >2.025} .  In Fig.3, the description on the right for GENEi+ was not accurate and  the description on the left for OPCML+, “Survival benefit with higher OPCML Expression”,   was not consistent with Fig. 1(a).   The interpretations of the results should be based on the ratio OPCML/GENEi.

  1. For each independent dataset, how many genes in the two gene sets OPCML+ and GENEi+ were validated? What were the percentages of genes in the two gene sets which were validated?  What were the KM p-values and HRs for the OPCML/CKS1BP3 ratio in these independent datasets?

Some minor comments:

  1. The meaning of ‘n=’ in line 87 is very different from that in line 178. To avoid confusion, it is better only to use ‘n=’ for a sample size.
  2. 6, line 229, where is Eq.2?
  3. In the legend for Table 1, ‘Figure 5’ should be ‘Figure A1’.

Author Response

Reviewer #1

Dear Reviewer:

We are grateful for the excellent insights and expertise provided through your comments. The errant details you picked up on have helped us improve the clarity of the manuscript. Incorporating all reviewers’ comments has resulted in a substantial revision that has increased the manuscript by 1 page to 18 pages. However, the corresponding increase in level of details has improved the messaging for a broader readership. Thank you.

MAJOR COMMENTS:

Major Comment #1:  The two gene sets were defined by OPCML+ = { Genei with HRz < -2.009} and GENEi+ = { Genei with HRz  >2.025} .  In Fig.3, the description on the right for GENEi+ was not accurate and  the description on the left for OPCML+, “Survival benefit with higher OPCML Expression”,  was not consistent with Fig. 1(a).   The interpretations of the results should be based on the ratio OPCML/GENEi.

Response MC #1: This is an excellent point you have made. The labels on the plot were done early on in the study. They no longer fit the original intention. Agree that the terminology needs to focus on the RATIO and we have modified the HRz text (p.7 lines 245-following):

            Functionally, these boundaries define two groups of genes: a) OPCML+: HRz values < -2.0 identify OPCML:genei ratios with a survival benefit under constitutive OPCML expression relative to genei levels suggesting that lower levels of genei expression may be linked to increasing efficacy of OPCML pathways of Tumor Suppression (i.e., better survival in the RNA-Seq High.Ratio group [see Fig. 2b]); b) GENE+

i : HRz values > +2.0 identify OPCML:genei ratios with a survival benefit that is not linked to OPCML expression and its pathways of Tumor Suppression (i.e., better survival in the RNA-Seq Low.Ratio group).

We completely agree that the labels were not consistent with the text. Figure 3 has been re-drafted without the definitions based on asserting survival benefit. The text in Results presenting HRz has been edited to reflect the focus on describing the ratio effect.

Major Comment #2:  For each independent dataset, how many genes in the two gene sets OPCML+ and GENEi+ were validated? What were the percentages of genes in the two gene sets which were validated?  What were the KM p-values and HRs for the OPCML/CKS1BP3 ratio in these independent datasets?

Response MC #2:  For our study, we selected the TCGA OV cohort because we could filter the almost 600 subjects down to 189 subjects specifically with Stage III ovarian cancer. We are confident that our results apply to Stage III ovarian cancer tumors. For the validation work, we are not trying to repeat the analysis. No two patient cohorts are identical. Usually they are not close in terms of the sources of variance within each cohort (age, race, stage, diagnosis, prior treatments, sample measurements). The inclusion/exclusion criteria in a study are tuned to the goals of the research team(s) and the population of patients available near enrollment centers.

When we extended the study to validate whether we can see similar OPCML:gene[i] relationships in other cohorts, we selected some of the top gene[i] features we identified from Table 2 & Figure 6. The goal was to assess whether key features identified in Stage III OVca cases were present in other studies of different OVca stages, subjects, data collection methods (GEO and TCGA RNA-Seq employ different types of assays), research projects, etc. Thus, the validation was focused on key features. The genes listed in the validation Table A1 represent about half of the total number of genes we tested, with those that were not validated omitted from the table. The appearance of KIT and TEK RTK transcriptional linkages in another cohort are substantial corroborative evidence of our methodological approach revealing true biological phenomena.

CKS1BP3:  Excellent question because it is highlighted as a methodological example in Figs 1&2. It was not on our validation list, but it is now: HR = 0.51, p-value=0.0099, cohort GSE14764 (c.f. in Fig2. HR = 0.657).

            This is a very important point to discuss and ensure reader clarity. We have expanded the section 3.4 Validation to include more details. CKS1BP3 has been added to the Validation table A1.

MINOR COMMENTS:

Comment #1:  The meaning of ‘n=’ in line 87 is very different from that in line 178. To avoid confusion, it is better only to use ‘n=’ for a sample size

Response #1:  Very good observation. We have different levels of sample groups and the use of n is not consistent

            We have revised text to specifically describe the numbers when we refer to them: 189 subjects, 60,600 features, etc. 

Comment #2:  line 229, where is Eq.2?.

Response #2:  Good catch. Vestigial reference from prior version where we presented the Z standard normal transform, however, given that this is “standard” we dropped the equation (and the three lines it occupied) to save space in the manuscript.

             Reference to Eq. 2 deleted.

Comment #3:  In the legend for Table 1, ‘Figure 5’ should be ‘Figure A1’

Response #3: Figure number has been revised to reflect current location in Appendix.

Reviewer 2 Report

Comments and Suggestions for Authors

Adam G. et al. This study analyzed the expression pattern and functional pathways of OPCML (a member of the IgLON family tumor suppressor gene) in ovarian cancer and revealed that it exerts multifunctional and pleiotropic tumor suppression by regulating receptor tyrosine kinase signaling pathways (e.g., PI3K/AKT and MEK/ERK) and transcriptional links to new potential RTKs such as KIT, TEK, and ROS1. However, I have several suggestions to strengthen the manuscript:

  1. How does OPCML affect multiple RTK signaling pathways simultaneously? Is it through a common mechanism or is it specifically regulated by different RTKs?
  2. What are the methodological advantages of using RNA-Seq ratios rather than single gene expression for analysis in the paper and what are the advantages of this approach over traditional differential expression analysis?
  3. Why is OPCML silenced in 82% of ovarian cancers and by which mechanisms this silencing is primarily achieved?
  4. In terms of clinical application, how can OPCML and its related pathways be used to develop therapeutic strategies against drug resistance?
  5. In the independent validation cohort, why have only some genes been validated? Does this suggest that certain associations may be specific to the TCGA dataset?
  6. What are the functions of each of the three Ig domains of OPCML and how do they work in concert to achieve tumor suppression?
  7. The paper mentions that OPCML impacts RTK signaling through lipid raft microdomains. How does this mechanism differ from other tumor suppressor genes, and what are the implications for targeted therapy development?

Author Response

Reviewer #2

Dear Reviewer:

We are grateful for the excellent insights and expertise provided through your comments. Sections of the manuscript have been expanded to better inform a reader relative to the points raised below. You raised excellent points of clarification that we believe improve the readability of the manuscript. Incorporating all reviewers’ comments has resulted in a substantial revision that has increased the manuscript by 1 page to 18 pages. However, the corresponding increase in level of details has improved the messaging for a broader readership.

Thank you.

COMMENTS:

Comment #1:  How does OPCML affect multiple RTK signaling pathways simultaneously? Is it through a common mechanism or is it specifically regulated by different RTKs?

Response #1:  A simultaneous regulation of multiple RTK pathways by OPCML occurs through distinct mechanisms reported in the literature. For example, OPCML binds to HER2 and promotes its redistribution into lipid rafts which promotes subsequent proteasomal degradation of HER2, thus reducing HER2-driven oncogenic signaling. In the case of AXL RTK, its sequestration by OPCML into lipid rafts is GAS-6 ligand-dependent and this triggers a lipid domain-resident phosphatase to dephosphorylate AXL. Furthermore, other distinct mechanisms exist involving OPCML-RTK interactions as a key node in broader signaling networks involved in tumor progression and metastasis. These include OPCML inhibition of EMT (e.g. in colorectal cancer) through increasing E-cadherin and suppressing TGF-β/SMAD signaling and vimentin levels.

            We have revised the text in paragraph 2 of the Introduction section to make this clearer to a reader.

Comment #2 : What are the methodological advantages of (a) using RNA-Seq ratios rather than single gene expression for analysis in the paper and (b) what are the advantages of this approach over traditional differential expression analysis?

Response #2: 

  1. a) RNA-Seq ratios – one common method in single expression analyses is to use a normalization gene (e.g., b-actin) to scale mRNA count data for all features. In this paper, we are selecting the “normalization” gene as OPCML because we want to assess variations in expression of other RNA features relative to OPCML.
  2. b) In contrast to Differential Gene Expression approaches that commonly use pair-wise contrasts in direct statistical comparisons (e.g. LRT-ANOVA models), in this work we utilize survival curve analyses to assess whether the RNA-Seq ratio with OPCML reveals a statistically significant impact on patient survival when the OV cohort is stratified by median ratio value per gene. The evaluation of the importance of the ratio of OPCML:gene[i] is based on a phenotype test (days_to_death) rather than just the statistical comparison of two RNA features.

            We have revised the Approach section in the M&M to include these differential points.

Comment #3: Why is OPCML silenced in 82% of ovarian cancers and by which mechanisms this silencing is primarily achieved?

Response #3: OPCML silencing in majority of ovarian cancers occurs primarily through epigenetic mechanisms, specifically somatic promoter hypermethylation. This means that the CpG islands in the promoter region of the OPCML gene become aberrantly methylated as an early step in ovarian oncogenesis, which prevents transcription factors from binding and thereby represses gene expression. To confirm this, demethylating agents such as 5-aza-2′-deoxycytidine have been shown to reactivate OPCML expression.

            The Sellar 2003 Nature Genetics paper is a good source for readers to consult. We have avoided discussing epigenetic mechanisms in any depth in the text to keep the scope of the paper manageable.

Comment #4:  In terms of clinical application, how can OPCML and its related pathways be used to develop therapeutic strategies against drug resistance?

Response #4: As a master regulator of RTK activity through non-canonical, extracellular mechanisms, OPCML offers several promising therapeutic avenues to combat drug resistance in cancer. These include its promotion of RTK degradation (e.g. HER2, FGFR1) or facilitating phosphatase-mediated deactivation (e.g. AXL). In these ways, OPCML restores balance to hyperactive signalling pathways commonly implicated in primary oncogenic signalling and also signalling associated with targeted therapy resistance. One strategy involves using recombinant OPCML protein therapeutically to reintroduce its tumor-suppressive effects through binding the ECD of RTKs at the external leaflet of the cell membrane within the lipid raft. Preclinical studies in ovarian and breast cancer models have demonstrated that treatment with recombinant OPCML reduces tumour growth and enhances sensitivity to small molecule RTK inhibition. Similarly, OPCML's repression of AXL, also in synergy with selective inhibitors, can more effectively target EMT in cancer cells especially as an approach to overcome resistance to prior chemotherapies. This includes targeting those cells’ enhanced invasiveness and metastasis, two hallmarks of aggressive, drug-resistant tumours.

            Very good question and we have added a drug-resistance paragraph to the section on Translational Application of Results, under Therapeutics.

Comment #5:  In the independent validation cohort, why have only some genes been validated? Does this suggest that certain associations may be specific to the TCGA dataset?

Response #5:  No two patient cohorts are identical. Usually they are not close in terms of the sources of variance within each cohort. The inclusion/exclusion criteria in a study are tuned to the goals of the research team(s) and the population of patients available near enrollment centers. We selected the TCGA OV cohort because we could stratify the almost 600 subjects down to 189 subjects with Stage III ovarian cancer. The unifying phenotype of our study is that all subjects have the same OVca development state. We are confident that are results are directly applicable to OVca Stage III cases. When we extend the study to validate whether we can see similar OPCML:gene$_i$ relationships in other cohorts, we selected some of the top features we identified from Figure 6. The goal was not to repeat the study in another cohort set, but to confirm that the most significant features we identified in Stage III cases are likely to appear in other studies of different OVca stages, subjects, data collection methods (GEO and TCGA RNA-Seq employ different types of assays), research projects, etc. Thus, the validation was focused on key features. The appearance of KIT and TEK RTK transcriptional linkages in another cohort are substantial corroborative evidence of the methodological approach revealing true biological phenomena.   

            This is a very important point to discuss and ensure reader clarity. We have expanded the section 3.4 Validation to include more details.

Comment #6:  What are the functions of each of the three Ig domains of OPCML and how do they work in concert to achieve tumor suppression?

Response #6: OPCML typically exists as a homodimer formed through interaction between Domain 1 (D1) of two molecules. Furthermore, D1 has been shown to be pivotal for binding to the extracellular regions of specific RTKs and mutations in D1 such as R65L disrupt these RTK interactions and impair OPCML’s tumor suppressive effects. D2 and D3 are clearly demonstrated to have additional important tumor suppressor function demonstrated through targeted mutation of critical amino acid residues as defined by clinically occurring missense mutations and this is fully described in our Birtley et al Nature Comms paper. Furthermore, many of the protein interaction partners are themselves composed of multiple sequential domains, including Ig, highlighting the essential roles of Ig domains in protein-protein interactions.

            There is nothing we can add to the manuscript that isn’t covered in the existing summary of Britley et al. work to assess impacts of point mutations in different Ig domains of OPCML.

Comment #7: The paper mentions that OPCML impacts RTK signaling through lipid raft microdomains. How does this mechanism differ from other tumor suppressor genes, and what are the implications for targeted therapy development?

Response #7: OPCML is unique in exerting its tumour suppressive function through RTK modulation through interaction with the RTK ECDs (as described above) in lipid raft microdomains. OPCML operates extracellularly as a GPI-anchored protein and its post-translational mechanism is clearly distinct from that of other defined tumour suppressors, including genes that function through transcriptional repression (e.g. TP53), chromatin remodelling (e.g. ARID1A) or the cytoplasmic protein PTEN which regulates Akt signalling.

This membrane-based mechanism has critical implications for targeted therapy development. First, it highlights the therapeutic potential of restoring or mimicking OPCML function, such as through recombinant OPCML protein delivery. Second, because OPCML can disrupt RTK dimerization and sensitize tumors to RTK inhibitors, its presence or absence could serve as a predictive biomarker for treatment response. Finally, unlike intracellular tumour suppressors that are often difficult to replace pharmacologically, OPCML’s extracellular localisation offers a more accessible therapeutic targeting approach for drug delivery.

            This is an important distinction and we have added another paragraph to section 4.3 Translational Application of Results.
